# Nanobiotechnology with Therapeutically Relevant Macromolecules from Animal Venoms: Venoms, Toxins, and Antimicrobial Peptides

**DOI:** 10.3390/pharmaceutics14050891

**Published:** 2022-04-19

**Authors:** Cesar Augusto Roque-Borda, Marcos William de Lima Gualque, Fauller Henrique da Fonseca, Fernando Rogério Pavan, Norival Alves Santos-Filho

**Affiliations:** 1Tuberculosis Research Laboratory, School of Pharmaceutical Sciences, São Paulo State University (UNESP), Araraquara 14800-903, Brazil; cesar.roque@unesp.br (C.A.R.-B.); fernando.pavan@unesp.br (F.R.P.); 2Proteomics Laboratory, School of Pharmaceutical Sciences, São Paulo State University (UNESP), Araraquara 14800-903, Brazil; mw.gualque@unesp.br; 3Department of Biochemistry and Organic Chemistry, Chemistry Institute, São Paulo State University (UNESP), Araraquara 14800-903, Brazil; fauller.henrique@unesp.br

**Keywords:** nanotechnology, venom, nanoparticles, drug delivery, drug discovery

## Abstract

Some diseases of uncontrolled proliferation such as cancer, as well as infectious diseases, are the main cause of death in the world, and their causative agents have rapidly developed resistance to the various existing treatments, making them even more dangerous. Thereby, the discovery of new therapeutic agents is a challenge promoted by the World Health Organization (WHO). Biomacromolecules, isolated or synthesized from a natural template, have therapeutic properties which have not yet been fully studied, and represent an unexplored potential in the search for new drugs. These substances, starting from conglomerates of proteins and other substances such as animal venoms, or from minor substances such as bioactive peptides, help fight diseases or counteract harmful effects. The high effectiveness of these biomacromolecules makes them promising substances for obtaining new drugs; however, their low bioavailability or stability in biological systems is a challenge to be overcome in the coming years with the help of nanotechnology. The objective of this review article is to describe the relationship between the structure and function of biomacromolecules of animal origin that have applications already described using nanotechnology and targeted delivery.

## 1. Introduction

Biological diversity has proven to be a rich source of many molecules with therapeutic potential and a solution to different public health problems [1]. Some animal species possess or produce biomacromolecules. For example, some venoms can act as a defense mechanism or contribute to immobilizing, killing, and digesting the prey [2]. In recent years, the rate of registration of new drugs at the Food and Drug Administration (USFDA) has declined, despite the concern issued by the World Health Organization (WHO) due to the emergence of new resistant bacterial strains and diseases resistant to therapy drugs [3,4].

Animal venoms constitute a vast source of bioactive molecules with pharmacological properties which have evolved for millions of years to efficiently interfere with the essential physiological processes of the prey [5,6]. Due to their toxicity and general biological effects, these natural resources have long attracted scientific interest [7]. Animal venoms are complex mixtures of peptides, proteins, salts, neurotransmitters, nucleosides, and other compounds [8]. Approximately 90% of the dry weight of some venoms (like that of snakes) consists of proteins and peptides. Among the protein classes, there are phospholipases A2, proteases (serine and metallo), l-amino acid oxidase, hyaluronidases, cysteine-rich secretory proteins (CRISPs), phosphodiesterases, nerve growth factors (NGFs), etc. Among the bioactive peptide classes, there are antimicrobial, analgesic, hypotensive, cytolytic/cytotoxic, cell-penetrating, anticancer (antitumoral) peptides, neuropeptides, and others.

The advances in the search for active biomolecules based on animal venoms indicates their effectiveness, which is mainly against cancer and diseases caused by microorganisms. [9,10,11]. However, it is well known that biomacromolecules from animal venoms, mainly proteinaceous ones, have low bioavailability or stability in biological media or in vivo treatments. In this context, few studies have reported the structure-function relationship of these biomacromolecules and drug delivery strategies [12].

Nanobiotechnology is a modern tool that helps to protect these compounds and even directs the target destination of the biomacromolecules, enhancing their activity and effectiveness [13]. Here, the structure-function relationship of biomacromolecules was studied; current applications and incorporation into nanostructured systems as a potential tool in the treatment of emerging diseases.

## 2. Nanosystems and Nanocarriers

Nanobiotechnology combines two important branches of modern science, namely nanotechnology and biotechnology. On the one hand, nanotechnology satisfactorily meets the physicochemical characteristics of protection and drug delivery. Its particles are on a nanoscale (size around 1–1000) and have pharmacological properties of high interest worldwide [14]. This is because nanocarriers optimize bioavailability, reduce adverse effects, and increase drug activity [15]. On the other hand, biotechnology is a science that studies the application or management of a living organism to improve or obtain a beneficial product [16]. This technological combination helps to obtain new compounds with high prospects on the market and benefits for the population [17].

In recent years, the macromolecules of animal venoms have been reported to have outstanding effects, such as antivenom, anticancer, and anti-inflammatory properties, among others [18,19,20,21]. However, this potential and activity are often limited by the different factors of the biological environment to which they are exposed, and many times these macromolecules lose activity or diminish their effect. For example, some proteolytic enzymes present in the blood may denature or break peptide and protein bonds.

Snake venoms are a complex mixture of biomacromolecules, containing different components, with more than 90% of their dry weight corresponding to proteins. Among the protein content of these venoms, there is a wide range of enzymes and peptides [22]. These molecules are responsible, alone or in synergism, for the physiological effects observed after animal envenomation. Venoms have been applied as therapeutics since antiquity, and technological advances have allowed the isolation and determination of the structure and properties of their components. As a result, their potential as new medicines has been explored in many fields. Macromolecules from animal venoms are considered high-profile biotechnological products and, combined with the protection of their components conferred by nanotechnology, highly impressive results are expected in the coming years. An overview of nanotechnology with these macromolecules and their strategies is presented below. In addition, controlled drug delivery systems (CDDS) in venom-derived biomacromolecules, such as nanoliposomes, nanoemulsions, polymeric nanoparticles, microparticles, microspheres, and mesoporous silica nanoparticles (MSNs), among others, are taken into consideration (see Table 1).

### 2.1. Nanoliposomes

Nanoliposomes (NLPs) are drug delivery and carrying systems made up of phospholipidic layers that cover an aqueous fraction [60]. NLPs have many advantages, but their main importance lies in their high biocompatibility, friendliness to the environment, and non-toxicity [61]. NLPs are vesicles that can be classified and distinguished depending on the method of preparation and size. They can be unilamellar (small unilamellar vesicles—SUV, 20–80 nm; large LUV, 80 nm–1 μm; and giant GUV, >1 μm), multilamellar (MLV, >400 nm), or multivesicular (MVV, >1 μm) [62]. Vesicle preparation requires careful formulation since the size of the particle differs when performed randomly. It was previously reported that the thin film hydration method would be an adequate formulation for antimicrobial peptides (AMPs) such as bacteriocins. This technique develops heterogeneous-MLV and, with assisted energy, it is possible to obtain LUVs or SUVs [63].

NLPs encapsulated with AMPs such as nisin, a cationic peptide, could confer, through its electrostatic properties, greater affinity for anionic lipids such as dicetylphosphate, or, through a hydrophobic affinity, zwitterionic neutral lipids such as phosphatidylcholine. Therefore, a better encapsulation efficiency is achieved in relation to anionic phospholipids such as phosphatidylglycerol [64,65,66,67]. Depending on the lipid and the electrostatic effect produced by the AMP, the stability of the NLP could be altered, and consequently better stability and zeta potential higher than 30mV would be achieved. The antimicrobial activity could also be related to the net charge of the lipid system formed, since the charge of the bacterial membrane is negative, which could repel the accommodation of NLP in pathogenic bacteria; this detail must be taken into account when formulating new antimicrobial agents [68].

Another peptide, Vicrostatin (VCN), has been encapsulated with NLPs. VCN is a peptide originating from the fusion between echistatin (*Echis carinatus*) and contortrostatin (*Agkistrodon contortrix contortrix* venom) fractions [69,70]. Vicrostatin is described in the literature as a potential anti-angiogenic and pro-apoptotic tumor agent [71]. Liposomal formulation of VCN was reported in vivo in mouse breast cancer; the data demonstrated that lamellar VCN is well tolerated while exerting a significant delay in tumor growth and an increase in survival in treated animals. In addition, the results showed antitumor ability in human ovarian cancer, glioma, and prostate cancer [23,24].

The trapping of animal venoms has become relevant due to the condition of transport to specific cancer cells, and it was shown that this type of venom-NLP nanocomposite achieves better anticancer effects in various cell lines [72]. Considerable application advances are being made in venom-NLPs in the biopharmaceutical industry since NLPs are already in the preclinical and clinical phases, mainly for the treatment of cancer [25,71,72].

Specifically, regarding the encapsulation of components from snake venoms with NMPs, some peptides can be cited. Melittin is the main component of the venom of the European bee *Apis mellifera*. It is a 26-mer peptide, originating from a two-step enzymatic cleavage of a preproprotein. The first cleavage removes an *N*-terminal signal peptide, while the second cleavage removes an anionic sequence responsible for inactivating melittin through interaction with C-terminal cationic residues [73,74].

Melittin’s primary structure is amphipathic, having a more hydrophobic *N*-terminal half and a more hydrophilic C-terminal half. Its secondary structure varies according to its concentration in an aqueous solution: at low concentrations, it assumes a monomeric random coiled conformation, while at higher concentrations, it adopts an α-helical tetramer folding [75,76]. The adopted α-helix conformation is a helix-hinge-helix motif, with the first helix formed by residues—a hinge region formed by residues (glycine-X-proline motif), and the second helix formed by residues [77,78,79,80]. These molecular and structural characteristics are directly related to melittin binding and permeabilizing different cellular membranes [76].

Melittin has been shown to exert anti-cancer, anti-inflammatory, anti-diabetic, anti-infective, and adjuvant properties. One of the studies carried out evaluated the ability of melittin to induce apoptosis in leukemia cell lines of different origins, acute lymphoblastic leukemia (CCRF-CEM), and chronic myeloid leukemia (K-562). Cell viability methods, membrane potential, and apoptosis were confirmed using two different cytotoxicity assays with trypan blue dye, and the MTT test was performed to identify this potential. It was concluded that melittin can be incorporated into phospholipid bilayers, resulting in cell membrane disruption leading to the intensive cytotoxicity of all treated cells. It was also observed that the apoptosis process is performed by the intrinsic/mitochondrial pathway in different leukemia models [81]. The ability to prevent biofilm formation was also found [82]. A recent report indicated that lipid-based NPs loaded with melittin exhibited modulative and targeted delivery properties, triggering the activation of hepatic sinusoidal endothelial cells, causing significant changes in the production of cytokines/chemokines in the liver, and allowing melittin to prevent the formation of metastatic lesions [26]. At the same time, melittin-NLPs, combined with a surfactant (2% poloxamer 188), showed high human hepatocellular carcinoma-toxicity: LM-3 (0.94 μM; 2.13 μM), Bel-7402 (0.90 μM; 1.44 μM), SMMC-7721 (1.13 μM; 2.03 μM) and HepG2 (0.66 μM; 2.19 μM). Melittin-NLPs showed anti-hepatocellular carcinoma (HCC) potency in vitro and in vivo with decreased toxicity, which has clinical implications as a promising new drug for HCC therapy [25].

Nanotechnology and intratumoral injection greatly decrease the required dose of melittin peptide, which generates cost reductions and significantly decreases its cytotoxicity, since melittin by itself is highly toxic and an uncontrolled dose could cause irreparable damage. A study revealed that α-melittin-NPs resulted in delayed tumor growth and even in the complete regression of the distant tumor. Furthermore, α-melittin-NPs have several strengths, including simple preparation, good stability, and a lack of side effects [27].

Finally, Alyteserin-1c is a linear peptide first isolated from skin secretions of the midwife toad Alytes obstetricans [83]. Studies by Subasinghage et al. [84] and Aragon-Muriel et al. [85] showed that the β-sheet structure is predominant in this peptide in an aqueous solution, and a transition to the α-helix structure was observed in a membrane environment. One study examined the antibacterial results of Alyteserin-1c and its analog [E4K]Alyteserin-1c. The growth-inhibitory potency of [E4K]Alyteserin-1c against clinically relevant Gram-negative bacterial reference strains was between two and four times greater than that of the original peptide sequence [84]. Although it has potential antimicrobial activity, polymer-coated alyteserin-1c-NLP significantly increased the antibacterial activity of the peptide against Listeria monocytogenes at 0.06 μM, showing excellent MIC-values in comparison with conventional drugs [28].

### 2.2. Silica Nanoparticles (SiNPs)

SiNPs are nanosystems with high drug delivery efficiency that vary in size from 10 to 500 nm. These nanosystems are the most widely studied and scaled for drug administration. Non-porous SiNPs are a type of SiNPs, characterized mainly by their large surface area, easy functionalization, and biocompatibility. They are formulated using microemulsions, but given that this nanosystem has been scantly used for macromolecules based on animal venoms, it is not discussed in depth here. One of the systems best adapted to peptides, proteins, and DNA is MSNs, since their ordered and porous design allows a greater load of the bioactive compound for therapeutic application [86].

MSNs are nanoparticles ranging in size from 2 to 50 nm, generated mainly by the Stober method or its modifications, and they are widely used on account of their simple manufacture, low cost, and availability on an industrial scale [86,87]. They can be classified according to their nano-application as MSNs sustained drug delivery systems (MSNs-SDDSs) which exhibit deliberate drug release, responding automatically to the conditions of the medium or a concentration gradient; and as MSNs stimuli-responsive controlled drug delivery systems (MSNs-CDDSs), which instead release the drug at a controlled level in response to a physical or chemical stimulus [88]. There are various types of MSN formulations, but most of the reports used “Santa Barbara Amorphous-15” (SBA-15) systems. These systems are characterized by their particle and pore size, which in turn allows a better and controlled release of the drug [89].

SBA-15 nanoparticles have a non-toxic and adjuvant effect which modulates the immune system in mice by inducing the production of IgG2a and IgG1 isotypes. This is beneficial when it comes to anticancer, anti-inflammatory, or antimicrobial treatments, especially in people with compromised immune systems [90]. This type of biotechnological formulation using SBA-15 is generally carried out under acidic conditions and using a source of silica, like tetraethoxysilane (TEOS), in combination with an amphiphilic triblock copolymer such as Pluronic 123 composed of poly (ethylene oxide) (PEO) and poly (propylene oxide) (PPO), i.e., PEO-PPO-PEO triblock [91]. The latest studies report its effectiveness as a pH-selective, photodynamic, and thermosensitive drug administrator, which can be administered through the gastrointestinal tract or by the intravenous, intratumoral, subcutaneous, ophthalmic, nasal, and dermal routes [91,92].

There are several examples of encapsulation of molecules derived from animal sources with SiNPs. Exenatide (39-amino-acid peptide), a molecule present in the saliva of the lizard from the Gila monster (*Heloderma suspectum*), is very similar to the incretin hormone glucagon-like peptide-1 (GLP-1). Clinical studies and clinical experience with exenatide have shown a significant reduction in HbA1c, fasting and postprandial glucose and a marked reduction in body weight in patients with type 2 diabetes mellitus. Animal studies have shown an improvement in blood function, beta cells, and an increase in beta-cell mass after treatment with exenatide [93]. GLP-1R agonists are promising anti-obesogenic and anti-dyslipidemia drugs in the early stages of obesity, in which the integrity of the nervous system has not been affected [94]. Currently, exenatide is available commercially under the names Byetta™ and Bydureon™.

Exenatide was incorporated into PEG/PLGA-NPs modified with Fc intestinal receptors for oral delivery to improve hypoglycemic effects in mice. Fc-targeted nano-delivery systems show great promise for oral peptide/protein drug delivery [42]. Exenatide-SBA15 MSNs were administered as injectables and served as glycemic modulators, since this peptide is a glucagon-like peptide 1 receptor agonist that that caused much greater inhibition than the wild-type peptide, showing a high loading capacity (15% *w*/*w*), as well as an increase in bioavailability and life span in blood circulation [29].

Studies of BSA-15 MSNs loaded the venom, previously detoxified with Cobalt-60 irradiation, from the South American rattlesnake (*Crotalus durissus terrificus*) and Africanized bee (*Apis mellifera*). The studies showed that their application as venom-antivenom adjuvants induced the production of antibodies in sheep, making it possible to obtain serum at a low cost in comparison with the common serum made from equine [30]. In another study, the same snake venom was incorporated into non-porous SiNPs and MSNs in a stable manner with no significant differences from each other, indicating that the V proteins, as well as the venom enzymes, were joined by electrostatic or ionic forces to the nanosystem, maintaining their physicochemical properties [31]. The venom of the *Micrurus ibiboboca* snake also immersed within SBA-15 MSNs facilitated the modulation and stimulation of antibodies for the development of possible vaccines [32]. This type of nanosystem has been widely used, primarily to obtain antivenom serum [95].

The *Walterinnesia aegyptia* venom was initially reported to have antitumor activity. Subsequently, it was encapsulated in MSNs, and its potential against breast, human multiple myeloma, and prostate cancer cells was demonstrated, along with a reduction in adverse effects and improvements in chemosensitivity, bioavailability, and specificity in cancer cells [33,34,96]. The authors also confirmed that controlled doses of these MSNs increased the levels of ROS, hydroperoxides, and nitric oxide in these cell cultures, thus increasing the levels of chemokines and decreasing the surface expression of their related chemokine receptors CXCR3, CXCR4, CXCR5, and CXCR6. Furthermore, they found that MSNs potentially inhibited insulin-like growth factor 1 (mediated by breast cancer) and epidermal growth factor (EGF, mediated by prostate cancer), while they induced apoptosis by increasing the activity of caspase-3, -8, and -9 [97].

Crotoxin (CTX) (l-Arginine esterhydrolase) is a polypeptide and the main component of *C. durissus terrificus* snake venom. It has favorable characteristics to be used as an immunomodulatory drug [98]. In a study whose purpose was to observe anticoagulant and profibrinolytic activity in endothelial cells, Andrade et al. performed an in vitro test on HUVEC cells to determine this possible activity. CTX exerted anticoagulant and profibrinolytic action in the presence of LPS, decreasing the levels of vWF and t-PA and increasing the levels of the proteins C and PAI-1, thereby representing a potential tool against the development of thrombosis [99]. The antitumor activity has also been reported against glioma (GAMG and HCB151), and pancreatic (PSN-1 and PANC-1) cancer cells showed greater sensitivity with IC50 of <0.5, 4.1, 0.7, and <0.5 μg/mL, respectively [100].

Nanoparticles of BSA-15 MSNs loaded with CTX demonstrated that their oral administration in mice significantly reduced/increased the production of interleukin (IL) 6 and 17/10, respectively. The toxic effect of CTX was reduced and consequently the LD50 values rose, which implies a favorable option for the treatment of chronic pain and experimental autoimmune encephalomyelitis (EAE). It was also noted that this nanosystem helped to improve the efficacy of CTX to treat multiple sclerosis because it reduced the proliferation of peripheral T helper cells 17 that characterize autoimmune, inflammatory diseases and cancer [20]. Crotalphine is a 14 amino acid analgesic peptide that, like CTX, trapped in SBA-15 MSNs was capable of increasing its therapeutic action, attenuating the inflammation induced by EAE because it reduced the immunoreactivity of Egr-1 markers, microglia/macrophages and astrocytes [101]. This information indicates the nanobiotechnological potential of both macromolecules and their protection and transport system, against the treatment of neurodegenerative diseases, control of sensitivity, motor alterations, atrophy, and loss of motor function [101]. Another study revealed that CTX-SBA-15 MSNs have a long-lasting contraceptive effect, as this nanoencapsulated system releases the CTX polypeptide in a controlled manner, which could be beneficial for use as new oral or injectable contraceptives [36].

Hylin a1 is a peptide isolated from the skin of South American tree frogs, which, in addition to exhibiting antibacterial activity, showed antitumor results. This peptide encapsulated in MSNs was efficiently released in the presence of tumor cells, and the hemolytic effect of the peptide could also be contained [37]. The results were corroborated by microparticles loaded with the Ctx(Ile^21^)-Ha peptide, isolated from the skin of the Brazilian Cerrado frog *Hypsiboas albopunctatus*, which also had decreased hemolytic activity using alginate coated with hypromellose acetate/succinate and hydroxypropylmethyl cellulose. This result provides evidence of the potential of micro and nanoparticles [102]. Ctx(Ile^21^)-Ha is a ceratotoxin that has shown potent antimicrobial activity against resistant bacteria such as *Salmonella* Enteritidis [103], multi-drug resistant bacteria such as *Pseudomonas aeruginosa* and *Acinetobacter baumannii*, as well as other pathogenic bacteria and fungi of public interest [104]. Its application in alginate microparticles significantly decreased its hemolytic activity, increased its bioavailability, showed a preventive effect of systemic and intestinal infection and greater physicochemical stability during its journey in the gastrointestinal tract [105,106].

Furthermore, the biotechnological potential of snake macromolecules was extensively studied more than a decade ago, as described in this section. However, the nano-venom or nano-macromolecule relationship is still a subject that requires further study, as previous results showed a great therapeutic effect. Its application in formulations for cancer treatment was approved by the FDA and, in general, the formulations based on silica were considered “safe”. However, in this type of nanosystem, the particle size must be taken into account because it could be related to the bioavailability and the rate of excretion of the drug [88]. Likewise, controlling the negative effects of this nanosystem is a great challenge; these types of nanoparticles could induce autophagy of hippocampal neurons through oxidative stress through the AMPK/mTOR/P70S6K signaling pathway, altering the nervous system [107]. These NPs induce the expression of nitric oxide, intercellular adhesive molecule 1, and adhesive molecule 1 of vascular cells in the human umbilical vein, suggesting that SBA-15 would be toxic to vascular endothelial cells [108]. Further studies of SBA-15 MSNs must be carried out since their prolonged use could lead to a negative risk/benefit action if they are not administered carefully. The authors concluded that animal venoms for use as antidote/serum/vaccine could be excellent nanocarriers since their administration, in many cases, is carried out in a single dose and is not prolonged compared to the administration of conventional drugs.

### 2.3. Metallic Nanoparticles

Metallic nanoparticles (MNPs) are nanosystems that incorporate the use of metals to enhance the antimicrobial activity of different bioactive compounds, the best known of which are gold (AuNPs) and silver (AgNPs). MNPs can currently be obtained by various methods; however, it is considered that these methods may cause environmentally toxic effects because the metals are difficult to degrade. For this reason, green synthesis processes are recommended [109].

Some venom components from vipers, such as crotamine, a peptide composed of 42 amino acids, with 4.8 kDa and isoelectric point around 10.8, were first isolated from the venom of the Argentinean rattlesnake *C. durissus terrificus* [110]. It is a small basic myotoxin containing three disulfide bonds (Cys4-Cys36, Cys11-Cys30, and Cys18-Cys37) [111,112,113,114] which acts on sodium and potassium channels [115,116], exerting myotoxic effects. Crotamine induces microorganism death by peroxidation and lipid oxidation of target proteins, determined by substances reactive to thiobarbituric acid and sulfhydryl groups, respectively [117]. Antibacterial and antifungal activities were evaluated in ATCC strains and clinical isolates, showing promising results [10]. Likewise, antitumor treatment was confirmed after the successful treatment of a mouse model grafted with a subcutaneous melanoma tumor [118]. Other biological activities, such as anti-leishmanial, anti-helminthic, and antimalarial effects, were reported [119,120,121,122]. Crotamine-AuNPs via a PEG linker were synthesized (NP size 14.6 nm); in vitro assays confirmed the internalization of these nanoparticles into living HeLa cells. AuNPs are playing a progressively more significant role in multimodal and multifunctional molecular imaging to detect and treat diseases, such as early-stage cancer [38].

Also, viper venom-AuNPs (*Daboia russellii russellii*, size 30–50 nm) provided protection against venom-induced edema, hemorrhage, defibrination, organ toxicity, and inflammation in an animal model (50 µg/20 g mice, intravenous) [39]. Other NPs containing lyophilized snake venoms, such as *Bothrops jararacussu*, *Daboia russelii* and *Naja kaouthia* (TiO_2_NPs) venoms managed to manipulate the antivenom effect and were able to control bleeding, clotting, and inflammation, and even to prevent the death of the animal; likewise, bacteriological dissemination was highly controlled [40].

The list of high-priority bacteria published by the WHO indicates that studies should be carried out in *Klebsiella pneumoniae* (critical), as well as *Staphylococcus aureus* and *S. typhimurium* (high) [3]. For this reason, a multibacterial study was carried out using a peptide sequence (INLKAIAALVKKV) from wasp venom (*Vespa orientalis*) in AuNPs. The results demonstrated excellent properties: the AuNPs were found to be nontoxic, small, inexpensive, heat mediated, and environmentally friendly, and they could have extensive medical applications as well. Based on their physicochemical and morphological parameters, it was shown that AuNPs may be synthesized and that bacterial multiplication may be blocked successfully using the peptide isolated [41].

### 2.4. Polymeric Nanoparticles

Polymers are linear or branched conformations of covalently linked monomers [123]. The nanoparticles obtained from this material are relevant because the loading content of the bioactive is efficiently high, and in most cases, the use of surfactants is necessary since many of the drugs are usually hydrophobic [124]. The versatility of polymers allows them to adapt to the different conditions that a drug or compound needs to improve its action or bioavailability, as well as structural chemical modifications to carry out conjugations [125]. Many polymers have been used as a source of PNs; however, this review will focus on those most widely used for the administration of derivatives of animal venoms.

Chitosan nanoparticles (ChitNP) are nanosystems that have been used in formulations for treatments involving mucoadhesion. Chitosan is a biopolymer that has a positive charge, and an affinity is generated by an electrostatic bond, targeting the mucosa, which is negatively charged; in addition, it has biocompatible characteristics and the ability to permeate the intestinal barrier, which stimulates the passage of molecules through the paracellular and transcellular pathways [123]. Although this polymer has certain limitations due to its low solubility at pH > 7, it is considered a good releaser of active compounds in the duodenum for gastrointestinal formulations. In vitro and in vivo studies have shown a non-toxic response, but the results of clinical studies that fully corroborate its application are still lacking [126]. ChitNP showed promising antimicrobial as well as anti-biofilm properties and is used mainly for the development of new nanostructured products, as it can destabilize the bacterial membrane and inhibit protein synthesis and mRNA transcription [14].

Poly(lactic-co-glycolic acid) nanoparticles (PLGA-NPs) are currently highly studied nanosystems. PLGA is an FDA-approved polymer due to compound protective potential, biocompatibility, and biodegradability [127]. However, PLGA-NPs have not shown good drug delivery properties in specific cells/proteins since the release of the compound occurs in an uncontrolled manner. Generally, this type of nanoformulation is accompanied by coatings or conjugates that can help this type of targeting, such as polyethylene glycol (PEG), polydopamine (PDA), hyaluronic acid, and folic acid, among others [128,129]. In addition, the cost-benefit ratio of using this polymer does not make it attractive since PLGA is relatively expensive and the particle size is around 500 nm. However, some non-toxic chemical agents can increase encapsulation rates and the release period can be carried out for as long as one month, as with bee venom (Park, Min-Ho 2016). Hamzaoui and Laraba-Djebar [43] reported obtaining and trapping the venom of *Cerastes cerastes* (horned viper) and *Vipera lebetina* in PLGA-NPs used as a vaccine; these reptiles are native to North Africa and the Middle East, respectively, and have a high bite incidence rate in the region. These PLGA-NPs (EE 84.1%) were administered intranasally, generating antibodies and high Th2 values, which favored humoral immunity; the NPs provided venom tolerance of 6LD50 and 5LD50, respectively [43].

Orivel et al. [130] tested the antimicrobial properties of fractions from the *Pachycondyla goeldii* ant venom against *Staphylococcus aureus* 209P and *Escherichia coli* RL65, and the best active fraction contained 15 peptides which were named ponericins and classified into the groups G, W, and L, according to the first amino-terminal amino acid. Ponericin-G1 was one of these promising peptides, since circular dichroism results suggested that ponericins G presented a random coil in aqueous solution and adopted an α-helical structure. In addition, researchers suggested that ponericins G acts through a “carpet-like” mechanism due to its similarities with cecropins and dermaseptins. Although it was the least concentrated peptide in the crude venom, it showed a great range of action against Gram-positive and Gram-negative bacteria and yeast (*Saccharomyces cerevisiae* LMA 720), and insecticidal activity against the cricket *Acheta domesticus* [131]. Ponericin-G1 was successfully incorporated into PDA-PLGA nanofibers for treatment against *Staphylococcus aureus* and *Escherichia coli* at 250 μg mL^−1^ and 250 μg mL^−1^, respectively. PLGA nanofiber scaffolds loaded with basic fibroblast growth factor and ponericin G1 prepared by polydopamine modification developed in this study has excellent mechanical properties, hydrophilicity, and antibacterial activity. This scaffold material has appropriate morphological and surface properties that can meet the needs of skin tissue regeneration [44]. Ponericin-G1 encapsulated in PLGA/PLL also showed effectiveness against *Staphylococcus aureus* and *E. coli* at 300 μg mL^−1^; PLL surface modification and ponericin G1 can greatly improve the antimicrobial activity of the cell microcarrier and enhance the clinical application value of microcarriers [132].

Finally, another toxin isolated from the *Leiurus quinquestriatus* scorpion venom is chlorotoxin (CHTX), which is a neurotoxin composed of 36 aminoacids that was first isolated by DeBin et al. [133] and is described as a calcium channel blocker. Selective and specific binding of CHTX to glioma cells was demonstrated by immunochemical techniques, and radiolabeled CHTX was shown to bind only to cells in a glioma tumor of a mouse xenograft model [134,135]. CHTX-morusin-PLGA NPs exhibited excellent pharmacological properties against the treatment of glioblastoma or brain cancer with a mechanism directed towards tumor cells. Likewise, there is evidence that crossing the blood-brain barrier is a great challenge that NPs manage to meet [45].

On the other hand, alginate-based nanoparticles (AlgNP) have excellent properties as mucoadhesive and molecule transporters, since several formulations have been made to achieve difficult-to-treat antibacterial objectives, such as *Mycobacterium tuberculosis* [136]. The AlgNP is obtained by crosslinking using a positively charged solution, which is generally produced by chloride salts, giving rise to the formation of nanospheres, nanocapsules, or nanoaggregates [13]. In addition, many of the formulations use this electrostatic property to combine the alginate and chitosan (Alg/Chit) and thus achieve greater protective stability of the active compound, mainly when it comes to gastrointestinal formulations [12]. Unlike alginate, Chit-NPs are obtained using ionic crosslinking, covalent crosslinking, polyelectrolyte complexation, and self-assembly [137].

*Apis melifera* bee crude-lyophilized venom in Alg/Chit NPs was applied to pigs to evaluate and compare its antimicrobial effect with the respiratory syndrome virus vaccine; it was shown that these nanosystems are capable of promoting the increase of CD4 T lymphocytes, as well as reducing the viral rate in the lung and bronchial lymph node [46]. Likewise, the use of ChitNPs with bee venom was successful in eliminating cancer cells and could be an excellent option for the treatment of cervical carcinoma [48]. In addition, this venom using the same nanosystem showed induced apoptotic characteristics such as cell shrinkage, peripheral chromatin condensation, late-stage apoptotic bodies, nucleolus fragmentation, membrane blebs, a shrunken nucleus with some necrotic characteristics in the colon (Caco-2), larynx (HEp-2) and breast (MCF-7) cancer cells [47]. These promising results encouraged the search for new macromolecules such as melittin, which is currently in preclinical and clinical phase studies [138]. Melittin-based nanosystems were also highly studied as PNs, lipid NPs, metallic NPs, carbon NPs, and other nanocomposites, as described with greater emphasis and detail in Zhou et al. [139].

A fraction of the attenuated venom of the scorpion (*Androctonus australis hector*) was encapsulated in alginate nanoparticles. These nanosystems led to an appropriate immune response to develop an alternative vaccine. The specific IgG titer and IgG1/IgG2a isotype values showed a potentially high protective effect because the immunized rabbits did not register a mortality rate with values of 6 LD50. This result corroborates the immunomodulatory capacity of the venom, and the physicochemical stability that the polymer grants to the antigen to achieve its adjuvant effect [50]. 

Another macromolecule isolated from the same scorpion venom, known as Aah II toxin-ChitNPs, confirmed the modulation of the innate and systemic humoral immune system. The nanosystem favored cell viability, decreased cytotoxicity (10 LD50), and had an EE of 96.66% (values considered excellent). In addition, the toxin was released at 86% until the fifth day. Therefore, the authors concluded that this effect can maintain constant protection and help prevent the death of patients who suffer bites from this type of scorpion, mainly in the most susceptible areas [51]. Furthermore, this attenuated venom in AlgNPs was used in preclinical studies in older animals such as rabbits; an effective immunizing response and safety were observed in its administration as a vaccine, while the treatment barely caused reactogenicity [49].

In 1996, Simmaco et al. [140] isolated three peptides from the skin secretion of *Rana temporaria*, which showed similarity to peptides isolated from the *Vespa* species. One of these peptides was Temporin B (TB), a leucine-rich hydrophobic peptide [140,141]. Structural studies showed that temporin peptides tend to adopt an amphiphilic α-helical structure in hydrophobic environments. These properties are related to the membranolytic action of this peptide through a barrel-stave mechanism. Interestingly, the existence of a lysine residue in TB was proposed as the factor reducing its activity against Gram-negative bacteria [142]. Later studies revealed that this lower activity against Gram-negative bacteria was related to the formation of TB-oligomers, interfering in the diffusion to the cytoplasmic membrane [143,144]. Besides its antibacterial activity, several studies in the literature report TB’s antifungal and antiviral potential. One of these studies characterized and synthesized TB_KKG6K and d-Lys_TB_KKG6K analogs, the latter having the l-lysines replaced by the chiral counterpart d-lysines to improve their proteolytic stability. It was observed that both peptides inhibited the growth of several species of *Candida* spp. fungi, *S. aureus* and herpes simplex virus 1 (HSV-1) [145,146]. ChitNPs loaded with Temporin B (extracted from Grass frog, *Rana temporaria*) were optimized in such a way that their antimicrobial activity could eliminate MDR bacteria in different strains of *Staphylococcus epidermidis* using only 5 mg ChitNP/mL with a prolonged release of up to 15 days and a significant decrease in cytotoxicity [52]. Another single macromolecule from scorpion venom known as *Tityus stigmurus* Hypotensin (TistH) was encapsulated in ChitNPs, demonstrating good biocompatibility, bacterial kill, and anti-biofilm activity of the most virulent species of *Candida* sp. [53].

Viper venom lyophilized-ChitNPs from *Bothrops jararacussu* showed relatively small sizes (average size of 160 nm), EE of 70%, and antibacterial activity in *Escherichia coli* and *Staphylococcus aureus*. It was demonstrated that NPs have the ability to enhance antibacterial activity on Gram-positive strains and that the incorporation of this snake venom is promising for obtaining novel therapeutic agents [147]. *Bothrops jararaca* and *Bothrops erythromelas* are two species of snakes from South America that produce highly dangerous venom. Soares et al. [148] reported the synthesis of antivenom-ChitNPs formulations at different concentrations of the venom (size of NPs ~167 nm) and EE in the range 67.7–97.2. These results indicated that the use of ChitNPs is capable of reducing localized toxicity and inflammation and that the interaction between proteins and the nanosystem can induce an adequate immune response to counteract the harmful post-venom effect. Another study, using speckled cobra (*Naja naja oxiana*), reported that venom concentration directly affects EE and the carrying capacity of ChitNPs [55].

*Crotalus durissus cascavella* venom proteins were incorporated into ChitNPs (stable and slightly smooth spherical NPs, <160 nm, SE 90%) as antivenom. They exhibited immunoadjuvant properties and reduced the adverse effects of venom, maximizing the efficiency of conventionally used antivenoms [54]. The authors also reported that these NPs are more effective than aluminum hydroxide, which is used as a conventional immunoadjuvant, suggesting that these nanosystems can be used in immunotherapy [149]. Using synthetic PNs, other antivenoms were developed from other snakes, such as *Crotalus atrox*, *Bitis arietans*, *Bitis gabonica*, *Echis ocellatus*, and *Echis carinatus.* The inactivation of metalloproteinases could be achieved for all of the species studied, which is interesting for the development of new enzyme inhibitors that are effective against snake bites [56]. The authors recommended caution when trying to obtain nanoparticles of protein or peptide bioactive compounds, as some methods/time of preparation or some compounds of the preparation solutions could break or inactivate a peptide bond.

## 3. Nanobioconjugations

Nanosystems loaded with animal venoms tend to increase the therapeutic effect, while bioconjugation can further increase this effect [40,150]. The bioconjugation of nanoparticles is generated by embedding a molecule in another nanomolecule. For this purpose, ligand techniques can be applied, the most promising ones being click chemistry application techniques such as Diels-Alder. Some nanoconjugates that are reported to stimulate and redirect nanoparticles are addressed in this review.

The SBA-15 system loaded with melittin uses the effect known as trigger-responsive valves, and shows the opening of the silica-based porosity at pH 5.5, which is the pH of cancer cells. This system was conjugated with 3-mercaptopropyltrimethoxysilane (MPTS), then with 2,2-Bis (*N*-maleimidoethyloxy) propane (MK-Linker, acetal), and finally with melittin, when a Cys was added to the sequence (d-amino acids) of melittin at the *N*-terminus [92]. Melittin-large pore MSNs capped by electrostatic interaction with β-cyclodextrin (β-CD)-modified polyethyleneimine (PEICD) were synthesized. It was observed that the MSN/pore size ratio was 90–110/7–10 nm, offering this nanobioconjugate (with assembly and loaded nanosystem) efficacy in the controlled release of AMP and/or high and low molecular weight drugs. In addition, this product was able to reduce the formation of biofilm (between 20 and 30%) and eliminate 24 to 27% of the total bacteria. However, the authors also explained that when using melittin-MSN combined with NPs-Magnetics of MnFe_2_O_4_ loaded with ofloxacin, previously synthesized by thermal decomposition, they were able to reduce the biofilm formation of *P. aeruginosa* PAO1 cells by up to 97% and eliminate 100% of this bacterium [151]. This study was corroborated using an in vivo mouse model, which exhibited a high rate of serum cytokine production in the groups not treated with this combined nanosystem, which clearly shows the use of nanobioconjugates with synergistic effect (see Figure 1) [152].

Some toxins, such as NN-32 (from the Indian cobra venom, *Naja naja*), have remarkable anticancer activity, but its high cytotoxicity does not make it an attractive macromolecule. However, when it was conjugated with AuNPs, the cytotoxicity decreased considerably without affecting its antitumor activity; this property was evaluated by means of in vitro (lymphocytes and macrophages) and in vivo (mice) assays [150]. In breast cancer cells, NN-32 incorporated into AuNPs induced apoptosis and discontinued tumor proliferation [150]. In addition, Lycosin-I from Spider (*Lycosa singorensis*) functionalized in AuNPs showed specific antitumor activity as an excellent nanocarrier to obtain novel drugs using these toxins [58]. Maurocalcine from *Scorpio maurus palmatus* functionalized with AuNPs showed selectivity for three different cell lines such as human epithelial-like HeLa, MDA-MB-231 (human breast adenocarcinoma), and A431 (epidermoid carcinoma cell), which allows nanosystems to be adapted to antitumor activity [59].

## 4. Prospects and Challenges

It is incontestable that peptides from animal venoms have therapeutic potential [153]. According to Van Baelen et al. [154], there are about 200,000 species of venomous animals around the world with about 40 million toxins to be exploited. However, there are only 11 peptide rugs derived from animal venom on the market, tens are in clinical development, and hundreds of patents have been filed in this field [155]. In our view, the reason for the low number of molecules currently on the market is that peptides, in general, have some limitations that need to be circumvented, such as proteolytic degradation [153]. 

However, to increase the half-life of peptides, many strategies involving different formulations, forms of administration, and levels of chemical modification are possible. In this context, an example of a promising strategy to increase the stability of the peptide is the introduction of d-amino acids [156], since the target of this class of peptides is the molecular membrane, which does not require interaction with specific chirality. Modifications in the N- and C-terminus of the molecule, the use of unnatural amino acid residues, cyclization, dimerization, and dimeric peptides are also used as alternative approaches to improve stability and increase biological activity [153,157,158,159,160].

Nanostructures have been utilized as vehicles intended to protect the peptides from enzymatic degradation or denaturation and prevent their release until they reach the site of action, giving larger specificity of molecules towards their targets. Furthermore, nanoscale materials have a large specific surface area, flexible surface functionalization, and some physicochemical properties that could be exploited to improve the molecule druggability [161].

The stability, toxicity, and cost of some nanoparticles should also be better studied and exploited. Large-scale synthesis will reduce the high costs associated with the synthesis and production of nano-pharmaceuticals over conventional treatments. NP stability in blood and environment depends on the materials they are synthesized with. Thus, the search for biodegradable NPs should be encouraged. Larger NPs have been shown to have a larger cytotoxic effect than smaller NPs, potentially accumulating in organs. Studies on the half-life, pharmacokinetics, and pharmacodynamics of these nanocompounds must be performed. For example, how do we remove the metal from the organism treated with MNPs? A study published in 2007 by Sadauskas et al. [162], indicates that Kupffer cells are essential in the elimination of NPs from the body. This information was confirmed and corroborated by Poon et al. [163], who added that hepatic sinusoidal endothelial cells would also be involved in this elimination through the canonical hepatobiliary pathway, as shown in Figure 2.

Regarding the complexity of therapeutic protein and peptide encapsulation, the field of nanotechnology has effectively provided several nanoscale systems that have the required encapsulating properties, as shown in the present work. As cited above, venoms are a complex mixture of compounds, produced by a tissue or organ (venom gland) and introduced mechanically into the prey by the venomous animal, with a specialized apparatus [164]. However, only a few examples of treatments using therapeutic proteins from animal venoms have been successfully translated into clinical applications, which demonstrates that the great challenge for researchers today is the translation of basic science into applied science [165]. The search for proteins and peptides to be used as therapeutic agents to treat cancer or infections has increased dramatically, mainly because of the emergence of resistance. Several biologically active toxins have been purified, characterized, and reported daily in the scientific literature. However, there is a gap between the initial drug discovery phase and its use in a clinical study [8], and this gap remains the biggest obstacle to be overcome.

## 5. Conclusions

Nanotechnology is a powerful tool that serves to enhance the therapeutic effect, transport, protection, and controlled release of macromolecules with biological activity. Although it is true that most studies are focused on administration as an adjuvant, current studies show that nanobiotechnology has great advantages, mainly reducing adverse effects and extending the half-life of substances derived from animal venoms. However, further studies of these biomacromolecules are necessary, as few nanosystems have been approved by the FDA for use in the pharmaceutical industry.

## Figures and Tables

**Figure 1 pharmaceutics-14-00891-f001:**
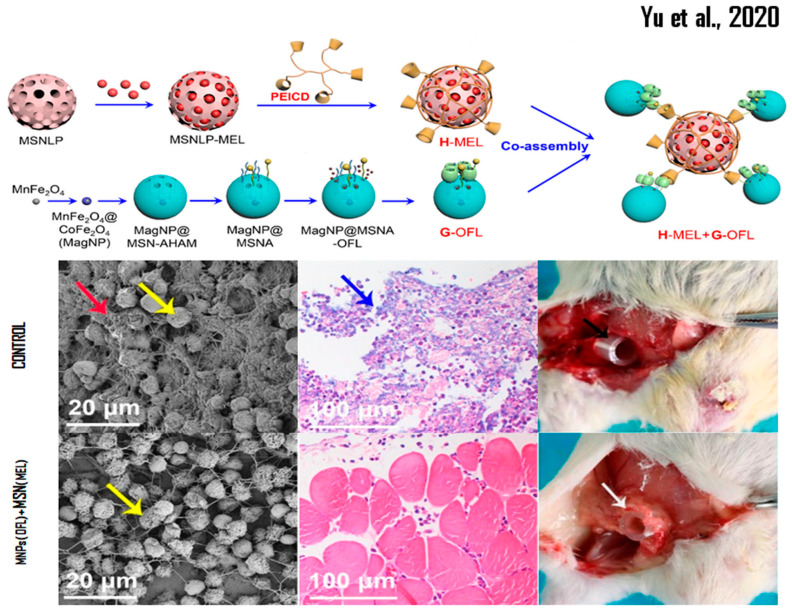
Representation of bioconjugated MSN nanosystems and their effectiveness in in vivo studies. Reprinted/adapted with permission from Yu et al. [152]. Copyright 2020, ACS Chemical Society.

**Figure 2 pharmaceutics-14-00891-f002:**
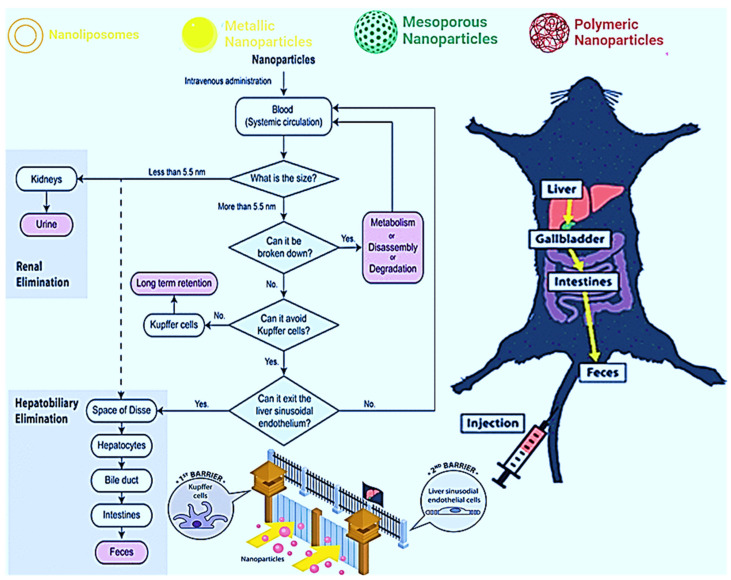
Schematization of the intravenous administration of NPs and the possible mechanisms of elimination: (1) liver, (2) gallbladder, (3) intestines, and finally (4) feces. Reprinted/adapted with permission from Poon et al. [163]. Copyright 2019, ACS Chemical Society.

**Table 1 pharmaceutics-14-00891-t001:** Animal toxins and nanotechnology: biomacromolecules, sources, nanosystems, and applications.

Toxin	Classification/Source	Nanosystem	Tested Activities	Refs.
Vicrostatin	Quimeric peptide Echistatin (*Echis carinatus*) and contortrostatin (*Agkistrodon contortrix contortrix*)	Liposomes	Anticancer (mouse breast cancer, human ovarian cancer, glioma, and prostate cancer)	[23,24]
Melittin	Peptide from *Apis mellifera* venom	Liposomes	Anti-hepatocellular carcinoma; Prevention of metastatic lesions	[25,26,27]
Alyteserin-1c	Peptide from *Alytes obstetricans* venom	Polymer-coated liposomes	Antibacterial activity against *Listeria monocytogenes*	[28]
Exenatide	Peptide from *Heloderma suspectum*	Mesoporous Silica NPs	Glycemic modulation	[29]
Detoxified venom	South American rattlesnake (*Crotalus durissus terrificus*) venom	Mesoporous Silica NPs Silica NPs	Modulation and stimulation of antibodies	[30,31]
Detoxified venom	*Apis mellifera* venom	Mesoporous Silica NPs	Modulation and stimulation of antibodies	[30]
Crude venom	*Micrurus ibiboboca* snake	Mesoporous Silica NPs	Modulation and stimulation of antibodies	[32]
Crude venom	*Walterinnesia aegyptia* venom	Mesoporous Silica NPs	Anti-cancer potential against breast cancer, human multiple myeloma, and prostate cancer cells	[33,34]
Crotoxin (CTX)	Beta-neurotoxin (PLA_2_) from *C. durissus terrificus* snake venom	Mesoporous Silica NPs	Potential activity against autoimmune, inflammatory diseases and cancer.	[35]
Crotoxin (CTX)	Beta-neurotoxin (PLA_2_) from *C. durissus terrificus* snake venom	Mesoporous Silica NPs	Potential contraceptive activity	[36]
Crotalphine	Peptide from *C. durissus terrificus* snake venom	Mesoporous Silica NPs	Potential activity against autoimmune and inflammatory diseases	[20]
Hylin a1	Peptide from South American tree frogs’ skin	Mesoporous Silica NPs	Anti-tumor activity and reduced hemolytic activity	[37]
Crotamine	Myotoxin from *C. durissus terrificus* snake venom	Gold NPs (PEG linker)	Anticancer and cellular imaging	[38]
Crude venom	*Daboia russellii russellii* venom	Gold NPs	Prevention of envenomation symptoms	[39]
Crude venom	*Bothrops jararacussu*, *Daboia russelii venom* and *Naja kaouthia* venom	TiO_2_ NPs	Prevention of envenomation symptoms	[40]
Peptide INLKAIAALVKKV	Peptide from wasp venom (*Vespa orientalis*)	Gold NPs	Antibacterial activity	[41]
Exenatide	Peptide from *Heloderma suspectum*	PEG/PLGA NPs	Glycemic modulation	[42]
Crude venom	*Cerastes cerastes* (horned viper) and *Vipera lebetina*	PLGA NPs	Development of an intranasal vaccine against envenomation	[43]
Ponericin-G1	Peptide from *Pachycondyla goeldii* ant venom	PDA-PLGA nanofibers	Antimicrobial activity	[44]
Chlorotoxin	Peptide from *Leiurus quinquestriatus* scorpion venom	Morusin-PLGA NPs	Anticancer activity	[45]
Crude-lyophilized venom	*Apis melifera* venom	Alginate/chitosan NPs	Antiviral activity and vaccine adjuvant	[46]
Crude-lyophilized venom	*Apis melifera* venom	Chitosan NPs	Activity against cervical carcinoma, larynx, and breast cancer cells	[47,48]
Attenuated venom	*Androctonus australis hector* scorpion venom	Alginate NPs	Development of a vaccine against envenomation	[49,50]
Aah II toxin	Peptide from *Androctonus australis hector* scorpion venom	Chitosan NP	Development of a vaccine against envenomation	[51]
Temporin B	Peptide from the skin secretion of *Rana temporaria*	Chitosan NPs	Antibacterial activity	[52]
Hypotensin	Peptide from *Tityus stigmurus* scorpion venom	Chitosan NPs	Antibacterial and antifungal activity	[53]
Crude-lyophilized venom	*Bothrops jararacussu* venom	Chitosan NPs	Antibacterial activity against Gram-positive bacterias	[54]
Crude-lyophilized venom	*Naja naja oxiana*, *Bothrops jararaca* and *Bothrops erythromelas*	Chitosan NPs	Prevention of envenmation symptoms	[55]
Venom proteins	*Crotalus durissus cascavella* venom	Chitosan NPs	Application as ant-venom	[54]
Crude venom	*Crotalus atrox*, *Bitis arietans*, *Bitis gabonica*, *Echis ocellatus*, and *Echis carinatus*	Polymeric NPs	Application as anti-venom, deactivation of metalloproteinases	[56]
NN-32	Peptide from the Indian cobra *Naja naja* venom	Functionalization of gold NPs	Antitumor activity	[57]
Lycosin-I	Peptide from *Lycosa singorensis* spider	Functionalization of gold NPs	Anticancer activity	[58]
Maurocalcine	Peptide from *Scorpio maurus palmatus*	Functionalization of gold NPs	Anticancer activity	[59]

NP: nanoparticle.

## Data Availability

Not applicable.

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
