# Peer review of "Nanobiotechnology with Therapeutically Relevant Macromolecules from Animal Venoms: Venoms, Toxins, and Antimicrobial Peptides"

_pharmaceutics, 2022, doi:10.3390/pharmaceutics14050891_

Round 1
Reviewer 1 Report
I have carefully read the manuscript entitled “Nanobiotechnology with macromolecules from animal venoms with potential therapeutic application: Venoms, Toxins and Antimicrobial Peptides” and find that it deserves publication, however, after some principal changes.
First, I would recommend to rethink the title to a less “bulky” one, for example, to “Nanobiotechnology with Therapeutically Relevant Macromolecules from Animal Venoms: Venoms, Toxins, and Antimicrobial Peptides”.
Second, I have certain difficulties upon reading the manuscript as it lacks structural clarity. Thus, the molecules from animal venoms which are discussed in view of their application in nanobiotechnology are neither summarized/divided into classes, nor even listed in a table containing the respective references. I would recommend to add a section between “Introduction” and “Nanosystems and Nanocarriers”. In this section, all venom molecules which are discussed in further sections (and rather randomly) must be mentioned and systematically assigned to certain groups depending on their chemical structure (peptide, protein, small molecule, etc.), or specific activity, or biological role, or by any other feature.
Third, it is very important to expand the section where the issues of stability and bioavailability are discussed. I would add a paragraph about “natural” ways to enhance stability of venom molecules, e.g. formation of multiple disulfide bridges or macrocyclization, and also to briefly summarize the “technological” ones. This will make a logical chain between the part discussing the therapeutic and the delivery counterparts of a nanosystem.
Another weakness (at least for me) is the lack of summarized arts of connection within discussed nanosystems: covalent bonds, ionic interactions, electrostatic forces, etc.
The literature is properly cited. However, I would add some recent reviews, e.g. Smallwood and Clark, “Advances in venom peptides drug discovery: where are we at and where are we heading?” Expert Opin. Drug Discov. 2021, 16, 1163-1173.
And a very minor point: letters D and L in the conformation of amino acids should be shown in small capitals.
Author Response
REVIEWER 1
I have carefully read the manuscript entitled “Nanobiotechnology with macromolecules from animal venoms with potential therapeutic application: Venoms, Toxins and Antimicrobial Peptides” and find that it deserves publication, however, after some principal changes.
First, I would recommend to rethink the title to a less “bulky” one, for example, to “Nanobiotechnology with Therapeutically Relevant Macromolecules from Animal Venoms: Venoms, Toxins, and Antimicrobial Peptides”.
We appreciate your comment, the title was modified.
Second, I have certain difficulties upon reading the manuscript as it lacks structural clarity. Thus, the molecules from animal venoms which are discussed in view of their application in nanobiotechnology are neither summarized/divided into classes, nor even listed in a table containing the respective references. I would recommend to add a section between “Introduction” and “Nanosystems and Nanocarriers”. In this section, all venom molecules which are discussed in further sections (and rather randomly) must be mentioned and systematically assigned to certain groups depending on their chemical structure (peptide, protein, small molecule, etc.), or specific activity, or biological role, or by any other feature.
We appreciate your comment, a paragraph and a table have been added between the introduction and the nanosystems. We also emphasize that the application and explanation of the macromolecules are ordered according to each nanosystem used.
Third, it is very important to expand the section where the issues of stability and bioavailability are discussed. I would add a paragraph about “natural” ways to enhance stability of venom molecules, e.g. formation of multiple disulfide bridges or macrocyclization, and also to briefly summarize the “technological” ones. This will make a logical chain between the part discussing the therapeutic and the delivery counterparts of a nanosystem.
Authors would like to thank you the suggestion and some information concerning the requested was added.
Another weakness (at least for me) is the lack of summarized arts of connection within discussed nanosystems: covalent bonds, ionic interactions, electrostatic forces, etc.
The main focus of this manuscript is the application of each nanosystem of protection and therapeutic action. For this reason, we believe that the discussion about the chemical structures and functions of nanoparticles can open up new points of view and perspectives that are not our main objective.
The literature is properly cited. However, I would add some recent reviews, e.g. Smallwood and Clark, “Advances in venom peptides drug discovery: where are we at and where are we heading?” Expert Opin. Drug Discov. 2021, 16, 1163-1173.
We appreciate your comment. The suggested reference was cited and discussed in the manuscript.
And a very minor point: letters D and L in the conformation of amino acids should be shown in small capitals.
We appreciate your comment. The D and L letters were modified in all manuscript.

Reviewer 2 Report
This review article is very exciting, as not much work have been conducted and addressed by the earlier researchers and covering the current nanobiotechnology and nanomedicinal applications of venom proteins and peptides, and thus well justified for consideration for publication. Despite aforementioned, it can be further improved drastically by the following major issues:
1) Extensive editing of English language and style required.
2) Figures provided are of low resolution and of not much relevance, thus MS rather required some better schematic illustrations to support the text, for example, adding some tables for quick reference to the readers instead of reading the whole text (i.e. summary tables of a) venom proteins and peptides and b) nanomaterials synthesized so far with info and applications).
3) Reference list, having a couple of duplications too.
Author Response
REVIEWER 2
This review article is very exciting, as not much work have been conducted and addressed by the earlier researchers and covering the current nanobiotechnology and nanomedicinal applications of venom proteins and peptides, and thus well justified for consideration for publication. Despite aforementioned, it can be further improved drastically by the following major issues:
1) Extensive editing of English language and style required.
Thank you for your comment. Consequently, we have sent the manuscript for English correction to a specialist.
2) Figures provided are of low resolution and of not much relevance, thus MS rather required some better schematic illustrations to support the text, for example, adding some tables for quick reference to the readers instead of reading the whole text (i.e. summary tables of a) venom proteins and peptides and b) nanomaterials synthesized so far with info and applications).
Thank you for your comment. Table 1 has been added, and the figures have been improved.
3) Reference list, having a couple of duplications too.
Thank you for your comment. All references have been carefully checked to avoid duplication.

Round 2
Reviewer 1 Report
Dear authors,
I read the revised manuscript and find it now suitable for publication.
Best Regards
Reviewer 2 Report
MS is improved significantly considering the both a) suggested table and figures and 2) english language editing and thus acceptable for publication.